# Historical Account of Managing Overabundant Wild Asian Elephants in Myanmar by the Kheddah System of Capture: Philosophy, Principles and Practices

**DOI:** 10.3390/ani14172506

**Published:** 2024-08-29

**Authors:** Khyne U. Mar

**Affiliations:** Asian Elephant Management, Asian Elephant Specialist Group (The International Union for Conservation of Nature), 12 Blackwell Place, Sheffield S2 5PX, South Yorkshire, UK; emaximus2014@gmail.com

**Keywords:** capture, *Elephas maximus*, human–elephant conflict, Kheddah, logging, Myanmar, Myanmar selection system, wild elephants

## Abstract

**Simple Summary:**

Historically, one of the strategies the Myanmar Government employed to resolve human–elephant conflict was the capture of whole herds of elephants using the Kheddah system. This involved trapping the herd in a stockade, immediately followed by on-site post-capture taming. After taming, the captured elephants were trained to be utilized as logging elephants. The capture of wild elephants was formally banned in Myanmar in 1985, but occasional, small-scale, captures by the Kheddah system were allowed until 2013. These captures were under the strict control of the Myanmar Government and focused primarily on elephants involved in human–elephant conflict, rather than capturing to supplement the working elephant population. The method itself has largely fallen out of use because capturing an ecologically important, iconic animal is unacceptable and against current welfare requirements. This paper attempts to record the Kheddah system of elephant capture that was employed when the individually selected capture method was impractical for the emergency removal of a herd of marauding elephants. The author presents a description and assessment of the Kheddah method not to endorse it, but to document and discuss a method that was part of Myanmar’s elephant-keeping history before it is forgotten.

**Abstract:**

When standard methods of human–elephant conflict mitigation are not successful, free-ranging wild elephants may continue to come into close contact with people. This results in more frequent and severe conflict, with consequences ranging from crop raiding to loss of human and elephant lives. Understandably, in such situations, local communities may want to be rid of entire herds of elephants. Historically, one of the strategies the Myanmar Government practiced to resolve human–elephant conflict was the capture of whole herds using the Kheddah system. This involved trapping the herd in a stockade, immediately followed by on-site post-capture taming. After taming, the captured elephants were utilized as logging elephants. Elephants worked in timber extraction, retired at age 55 years and were then cared for by the Government until they died. The capture of wild elephants by the Kheddah system was formally banned in Myanmar in 1985 but occasional, small-scale, captures were allowed until 2013 under the strict control of the Myanmar Government. These operations were aimed primarily at capturing elephants involved in human–elephant conflict, rather than to supplement the working elephant population. One of the last Kheddah operations, organized and managed by the author, was conducted in 1996 at the Taikkyi township of the Yangon Region, as a last resort to end human–elephant conflict in an emergency. While chemical immobilization was being used at this time, it was not logistically possible with the high numbers of elephants engaged in the conflict. This review aims to record the history of an activity that was an important element of Myanmar’s timber industry more than three decades ago. In this paper, the author presents a description of Kheddah not to endorse it, but to document (1) the Myanmar elephant population management strategy in the past, before it is forgotten, and (2) the practicality of the Kheddha operation when the singly selected commonly- used immobilization or noosing method of elephant capture is unfeasible. The author attempts to shed light on the modern veterinary procedures that may significantly reduce the notorious historical outcome of Kheddha, especially the resulting mortality of captured elephants, should the Kheddha system of capture ever be used as an emergency solution for ongoing problems of human–elephant conflict in the range states of Asia.

## 1. Introduction

The Asian elephant (*Elephas maximus*) is categorized as “endangered” in the IUCN Red List of Threatened Species [1] and listed in Appendix I of the Convention on International Trade in Endangered Species of Wild Fauna and Flora (CITES). They are recognized as a “keystone species” because their impact on an ecosystem or community is disproportionately large relative to their abundance, and they are key in maintaining the integrity of the ecosystems they inhabit [1,2,3]. They may also be considered “flagship species” because of their iconic or cultural value [1] and classed as “umbrella species” because of the requirement of a large habitat that will bring many other co-habitat species under protection [1,4,5]. Historically, Asian elephants have played a crucial role as beasts of burden, significantly contributing to the economies, military operations and cultures of the countries they inhabited, and elephant populations were much larger than they are today [6,7].

Myanmar has the second largest population of wild Asian elephants (after India) and the largest number of domesticated elephants in Asia [8,9]. Precise numbers of Asian elephants in the early 1900s are challenging to determine due to the lack of systematic surveys and reliable records from that period. It was estimated that there were tens of thousands of wild elephants in Burma in the early 1900s, especially in areas bordered with Assam, Bangladesh, Laos and Thailand [10]. Wild elephants were found throughout Burma, especially in Bago Yoma, Tanintharyi and Rakhine Yoma, as well as in the northern and western parts of the country [11], and the only places wild elephants could not occupy were places closed to dense human settlement areas [9]. The estimate of the wild elephant population of Burma between 1935 and 1962 was 9000–10,000 [10,12]. During this period, live capture from the wild was the main method of replenishment of the captive elephant work force that was engaged in logging operations. Despite the significant removals of wild elephants for logging operations, wild elephants thrived and the population remained high into the 1970s [13,14]. The Burmese Government’s estimate of the wild elephant population in the 1990s was ≈5000 [15,16,17]. A range-wide assessment of the remaining elephants in 2003 suggested that Myanmar/Burma still possessed a more high-quality habitat than neighboring countries [9,13].

Ongoing anthropogenic factors, such as the encroachment of forestland for subsistence farming and human habitation, illegal and unsustainable logging, massive establishment of forest plantations, development of infrastructures and local dependency on forest resources led to a considerable loss of elephant habitats in Myanmar [8,18]. When the habitat decreased and became more fragmented, wild elephants inhabiting some areas of Myanmar faced the same fate as in neighboring countries [13,18,19,20], leading to intense conflicts between people and elephants, causing fatalities on both sides besides damage to human property [1,21].

One of the strategies the Myanmar Government practiced to resolve the human-elephant conflict was the capture of whole herds using the Kheddah system, and the captives were trained to obey verbal commands and used as transport or logging elephants. Such operations were aimed primarily at capturing elephants involved in human–elephant conflict, rather than supplementing the working elephant population. The constant off-take of wild elephants to supplement the country’s working elephant population since the mid-1900s is blamed for the decline in the wild elephant population [8,9]. The author attempts to justify that wild capture using the Kheddah system can be regarded as a viable tool for the emergency removal of locally overabundant elephant populations in Myanmar and also as a genetic management method of restoring gene flow to a captive population.

## 2. Utilization of Captive Elephants in Myanmar

The Union of Myanmar (formerly known as Burma), with an area of 676,553 square kilometers, is one of the largest in mainland Southeast Asian countries. Elephant draught power has been utilized extensively for timber harvesting for more than a century [9,22,23]. In Myanmar, the State owns all lands, including forests. Under the Ministry of Natural Resources and Environmental Conservation (MONREC), the Forest Department (FD) is responsible for the management of forest resources, while the Myanmar Timber Enterprise (MTE) has a monopoly over the country’s timber extraction, processing and trade. The basic management principle for forests in Myanmar is known as the Myanmar Selection System (MSS) [24,25,26,27,28]. Under the MSS, forest lands are organized into felling series, each of which is divided into 30 blocks of an approximately equal yield capacity. The principal concept of the MSS is to maintain sustained yields without depleting the resource and causing minimal environmental degradation. The distinctive feature of the MSS is the harvest of marked trees of an exploitable size in a prescribed plot of forest. Skidding or dragging logs by elephants can fulfill the requirements of the MSS and have less of an impact on residual trees compared with machine-only operations [25,28]. A machine is used only when felled trees are too big for elephants to carry. Reduced impact logging (RIL) practices were developed in the late 1980s and early 1990s by a variety of forestry researchers and institutions aiming to mitigate the environmental impact of logging operations [29]. In this context, using elephants for logging operations is well-acknowledged as RIL because elephants do not require extensive road networks or heavy machinery, thus minimizing soil compaction and the disturbance to the forest floor [25,28].

There are ≈15,000 Asian elephants in captivity globally [19]. Myanmar is widely recognized as home to the largest captive. Asian elephant population (~6000), and the majority of captive elephants is used in the logging industry [9,11]. Around 200 are engaged as transport animals for tourism or religious ceremonies or as exhibit animals in zoological facilities [11]. It is generally considered that the State (in the form of the MTE) owns 50% of the elephants, while 50% are privately owned [11]. The precise demography of privately owned elephants is not available because privately owned elephants in remote parts of Myanmar cannot be registered with the Government registration scheme [11]. The number of elephants under the MTE’s management was reported as 3122 in December 2018 [30] and increased to 3287 in January 2023 (MTE data, 2023).

Mechanical extraction is exercised only in limited areas. The date, heavy equipment has been used mainly for road construction, loading and unloading logs and transportation [14,31]. Only half of the MTE elephant population (≈1600) can engage in harvesting operations at any one time, because young elephants (<18 y), pregnant females, females with calves (<5 y) classified as calves-at-heel (CAH), old elephants (>55 y) and sick elephants are not allowed to work in extraction operations [17]. The MTE hires elephants from private owners on a contract basis according to extraction quotas. The captive elephant population in Myanmar is characterized as semi-captive because they are released unsupervised into the forest at night to forage with their family groups, where they have access to a variety of natural vegetation [31,32]. This manuscript focuses on the Government-owned captive elephant population employed by the Myanmar Timber Enterprise (henceforth, MTE elephants or elephants). While Burma was under British rule (1885 to 1948), elephants engaged in logging operations were traditionally kept under an extensive keeping system where they were released at night [31,32]. The unsupervised free foraging at night enables the camp elephants to socialize and mate with wild elephants and vice versa because most timber camps are situated in the vicinity of forests where wild elephants roam [33]. Breeding is natural and not managed. This is one of the reasons that Myanmar elephants have maintained high levels of genetic diversity [34,35]. Assisted parturition is infrequent because females generally give birth in the forest. Dystocia (abnormal or difficult in parturition) has rarely been reported [33]. Humans do not intervene with the care of calves that receive maternal and allomaternal care until they reach the age of taming at four years old [33,36]. Females with suckling calves are relieved of logging work until the calf reaches one year [33]. Reproductive dysfunction due to long non-reproductive periods as seen in zoos [37] has not been reported to date in MTE elephants.

## 3. Capture of Wild Elephants

Historically, several methods were used to capture wild elephants in Asia. In Myanmar there were six: Kheddah (stockade); Milashikar (lassoing method); immobilization (using the drug etorphine hydrochloride); the decoy method (using female elephants as decoys); nooses (hidden snares) concealed on the ground and the pit method (trapping elephants in camouflaged drop-in pits). Among these methods, Myanmar mainly used Kheddah, Milashikar and immobilization methods to legally capture wild elephants to replenish the State-owned captive stock. Using the Milashikar or immobilization methods, only one wild elephant could be captured at a time. With the Kheddah system, a group of elephants of various sizes and sexes could be captured in a short period.

Specially trained tuskless males, known as Konchee elephants, were indispensable, especially for the Milashikar and immobilization methods, where they were used in the capturing operation. In the Milashikar and immobilization methods, capture was aimed at sub-adult (≈10 year) elephants so that the Konchee could guide the captives back to the base camp. Konchees were a special group of elephants selected for their huge bodies (average height at shoulder ≈ 2.7 m), bravery, intelligence, obedience and with numerous testimonies as proof for protecting the humans riding on their shoulders [38]. These males were specially trained to understand the signals used by the catchers, such as the mahouts’ leg pushing behind the ears, knee pressing on their heads, finger pressing on their necks, etc. These signals were employed during the phase of chasing and cornering the captured elephant, so that the elephant catchers could noose the legs or body parts, ready to drag the captive to base camp for on-site taming. Male elephants with tusks and female elephants with tushes (small tusks) were normally not involved in capture operations, but rather used as transport or baggage elephants to avoid the possibility of unintentional trauma to the captured elephants by the tusks and tushes during chasing, pushing, head-butting and cornering.

The Kheddah system of capture that was practiced before the 1900s involved operations where wild elephants were driven into a stockade by skilled oozies (mahouts) mounted on trained elephants. The word Kheddah is a Hindi word meaning the enclosure intended for imprisoning the herd of wild animals [39]. According to Shell (2019) [38], the holding area resembled a symbol for gamma (γ) with a huge ‘V’ shape and a tiny circular enclosure at the end, while Toke Gale (1971) [10] described the whole entrapment as shaped like a huge capital Y where the arms of the Y formed a wide funnel and the short leg of the Y was a narrow holding area or enclosure. The Kheddah system described here is different from the method practiced widely in Northeast India (particularly in the state of Assam) and South India (particularly in Mysore state, now part of Karnataka state) during the early 1900s, where the whole herd of wild elephants with mixed sexes and various age groups was driven into a football pitch-sized enclosure. The Myanmar Kheddah system was developed by private elephant catchers of Karen ethnic minorities (living in the middle and southern parts of Myanmar) who split the wild elephant herd into male and female groups during the drive; only the female group was led into the stockade for capture.

## 4. Physiology of Kheddah

Traditionally, the capture of wild elephants was strictly controlled by the Forest Department, under the direction of the Ministry of Forestry of the Government of Myanmar, which is responsible for the conservation and management of forests and for developing the Forest Management Plan, Regulations and Laws. The capture operations were conducted by the State Timber Corporation (the precursor of today’s Myanmar Timber Enterprise).

During early post-independence times (circa 1950s), the State Timber Corporation purchased elephants captured by approved professional elephant catchers who followed the existing regulations for elephant capture. Those who violated the existing scheme (such as capturing wild elephants in non-regulated areas, capturing more elephants than the numbers of animals set by the Forest Department, those with a history of high post-capture mortality of elephants, etc.) were black-listed and never allowed to capture wild elephants again. Since 1983, the immobilization technique has been the only method used for the capture of wild elephants in Myanmar, and all capture operations have been performed by MTE veterinarians [31].

It has been reported that the mean capture age by the Kheddah system was higher than that of the Melashikar or immobilization methods of capture [32]. Most elephants captured in stockades were females, including matriarchs, pregnant females, juveniles and mothers with suckling calves. Private elephant catchers tended to avoid males with tusks because adult males, fearful and enraged, were liable to injure or kill other elephants in the stockade. Males were also more difficult to capture and took a long time to tame.

According to government records, between 1952 and 2000, 2161 elephants were captured alive from the wild with an average rate of capture of 45 elephants per year; across all years between 1952 to 2000, 59% of captures were performed using the Kheddah system, 29% by immobilization, and 12% by the Melashikar method [32]. Records document that the sex ratio was strongly female-biased in the captured population [32]. The total number of wild-caught elephants gradually declined after the 1970s as captures declined (Figure 1).

High mortality during and immediately post-capture was often documented [15]. The post-capture mortality rate of elephants captured using the Kheddah system was 30.1% [15] while others reported a rate as high as 60% [40]. Deaths were partly due to the traditional breaking method practiced at that time (circa 1970s) coupled with the lack of qualified veterinarians in the workforce of the early administration to monitor the welfare and health of elephants that underwent taming and lax supervision by government officials during and immediately after capture [41]. Knowledge of basic animal welfare and systematic elephant training methodologies was nearly non-existent at that time. For these reasons, the use of the Kheddah system was officially banned in Myanmar in 1984.

## 5. Principles of Kheddah

The most important principle of the Kheddah system is keeping the captured animals as safe and injury-free as possible. In a Kheddah operation, a herd of elephants is selected and driven into a stockade for capture. The earliest detailed accounts of the Kheddah system are provided by Tennent (1867) [42] and Sanderson (1887) [43], and later by Shell (2019) [38]. Kheddah operations in Myanmar were detailed by Toke Gale (1971) [10], Zaw (1997) [31] and Mar (2007) [32]. Although the Kheddah system was officially banned in 1985, provision was made for the continued use of the Kheddah system as an emergency solution to human–elephant conflict until 2013. One of the last few Kheddah operations after the 1985 ban was conducted in 1996 in the Taikkyi townships of the Yangon Region, where the author was in charge of the entire operation. Six adult females (estimated age between 10 and 45 year) and a female calf (estimated age 1 year) were caught alive. Among these captured elephants, three adult animals (0M, 3F) were transferred to the MTE. The captured team received the remaining elephants as per the previous agreement with the MTE.

In Myanmar, a Kheddah team was traditionally composed of 30 men with a leader. The guidelines set by the Myanmar authorities emphasized that the primary focus of all capture events must be the safety of both the personnel and the captured elephants. High post-capture elephant mortality and accidents due to elephant attacks or escapes were regarded as unprofessional. The capture team was required to have achieved the following steps well before the capture.

The composition of the herd structure, especially the sex, age and number of adults, sub-adults and calves in the target herd, had to be known. Groups containing many older/mature males (40+ years) and pregnant females were not suitable for capture by Kheddah. Older elephants caught in the wild generally require a longer period of taming than younger elephants. Herds with sub-adult (estimated age of 10+ year) elephants were normally chosen by private/professional capturers.The routes and foraging locations of the target herd had to be monitored for six to eight months prior to capture.A site for the construction of a stockade had to be chosen. This was ideally on a flat terrain or a slightly sloping area with an abundance of shade, fodder and water. The area needed easy access for baggage elephants, allowing rations, medicine and construction equipment to be carried in without difficulty. The construction site of the stockade had to be spacious enough for dragging captured elephants by force from the stockade by manpower or, sometimes, with the help of Konchee (helper) elephants. The construction of the stockade needed to be carried out with minimal soil disturbance. There had to be sufficient flat land nearby for the taming of captured elephants.The driving path of the wild elephants toward the mouth of the stockade had to be far from ravines, rivers, slopes, cliffs and creeks. These features could impede the drive or act as traps for elephants or humans.Team members had to be familiar with, or trained in, capture methods, and aware of their advantages and drawbacks.To avoid hyper- and hypothermic stress due to extreme ambient conditions, which include extremes in air temperature, wind, solar radiation and/or precipitation, the procedure had to be conducted within a pre-defined safe temperature range. In practice, this usually meant in the early morning or late evening before dark.Mentoring by experienced persons was strongly recommended as the best approach to becoming proficient at capture. If and when possible, a security team of armed personnel from the police or army would be incorporated into the capture team. This was to prevent accidents caused by wild bull elephants possibly related to family groups of captured elephants, especially during the post-capture taming period.Capturing wild elephants was usually practiced in the cool season (October to January) and never during the monsoon and summer, due to practical difficulties. Captures were preferably aimed to be finished by December because of the need to finish the taming of captured elephants before summer set in in mid-FebruaryAll constructions and waste at the capture site needed to be removed or burnt after the end of taming sessions and the soil needed to be left in good condition to let vegetation grow naturally.

The traditional Kheddah system used in Myanmar was shaped like a huge capital Y [10]. The arms of the Y were two wooden railings, 350 to 550 m long, which formed a wide funnel (also known as the “wing”). The short leg of the Y was a narrow holding area or enclosure. The short leg was about 3.5 m wide at the mouth and gradually tapered off to a dead end, where the width was less than 2 m or a stout tree could be used as the top of a dead end. Such a holding area could hold up to 15 individuals. Figure 2a,b are schematic diagrams of a stockade and show the details of a stockade construction, as seen in Tennent (1867), including how vertical and horizontal posts were positioned to make elephant-proof fences.

The stockade was made of posts four to five meters in length, and/or standing trees of this height, 0.3 m or more in diameter. These posts were dug one meter into the ground and spaced at intervals of approximately 0.3 m. They were then interspersed and reinforced with a set of poles of similar sizes and fastened together horizontally with strong jute ropes or rope made from the fiber of the Sterculia tree. The horizontal posts were also placed at intervals of 0.3 m, from the ground level to the top. This formed a strong lattice of strapped poles (Figure 2b and Figure 3a). The lattice walls were reinforced from the outside by long poles, the butts of which were buried and the tops of which were braced against the lattice (see detail in Figure 2b and Figure 3a). On one side of the holding area of the stockade, an opening large enough for an elephant to pass through was created with an exit door that could be easily shut or opened (Figure 4). The stockades were made without a single nail. Posts, poles and ropes were sourced from the jungle. The last stage of stockade construction was to camouflage the whole entrapment area with green branches, banana trees, twigs and leaves (Figure 3a).

At the point where the long arms of the ‘Y’ joined, there was a heavy drop-gate (Figure 3b). This gate was made very strong using thick, straight poles and suspended some nine meters above the ground by means of a 15 cm thick jute rope or rope made of the fiber from the bark of the *Sterculia villosa* or *Sterculia versicolor* tree. This rope, which was 25 m long, was thrown over the beam that spanned the entrance, and then stretched taut over the entire length of the stockade and fastened around the stout tree that supported the narrow end. When this rope was severed, the drop-gate forcefully and swiftly closed the holding area. A small platform or bridge about 1 m^2^ was constructed on top of the entrance where the gate-keeper, usually the team leader, was stationed at the time of the drive (Figure 3c). At the right moment, he swiftly cut the rope using a very sharp knife. The construction of Kheddah described in this manuscript can be visualized in attached video documentary (Appendix A), showing that the method and the materials used in the construction were the same in the late 1800s [42].

## 6. Practice of Kheddah

Because of the damaging reputation of Kheddah-related mortality, private elephant catchers developed a new method of driving wild elephant herds by splitting the herd into two groups. Previously, the selected herd had been driven toward the stockade by the Kheddah team, and all elephants entering the stockade were caught and tamed regardless of sex or age. It was noticed that some mature elephants, particularly males, died during taming, or survived poorly in captivity. This was likely due to the elephants’ response to human presence along with the physical and psychological trauma sustained from capture. Anecdotal reports claimed that females captured by Kheddah were easier and quicker to tame. Private elephant capture teams preferred to catch young females, because of the chance for them to have a calf. This was one of the driving forces of wild off-take for private elephant catchers in the late 1960s. The catchers preferred to catch female-biased groups of wild elephants with an estimated age of 20 years. Full-grown bulls and old females were not desirable.

Once the target herd was located, it was gradually driven toward the stockade by the Kheddah beaters who shouted, blew bamboo horns or set off firecrackers. Elephants can detect sounds as low as 14 to 16 Hz (cf. human: 20 Hz) [44], so the driving sounds of the beaters and bamboo horns (typically somewhere between 90 and 130 decibels) accompanied by shots and/or fire were highly effective. Elephants rushed forward to escape the noise, apparently in a frenzy. The beaters’ main object was to keep the herd in front of them. To direct the elephant herd into the stockade and to prevent them turning back, members of the Kheddah team lit fires in piles of dried twigs, leaves and branches that had been collected and put in place before the drive. The first group of elephants to respond to these sounds and flames was composed of mostly young adult males. A second group, which responded later, was the matriarch-led adult female group, with accompanying sub-adults and calves. As they fled, the adult females also desperately tried to surround and protect the younger herd members.

Once the two groups, mostly male- and female-led, were clearly separated, some of the Kheddah team diverted the only male group by firecrackers, shooting into the air or shouting, preventing them from entering the stockade. Once the male group had been diverted, the female-led group was carefully directed to the funnel mouth of the stockade. The drive finished with an intensification of shouting, beating of bins and lighting of firecrackers. When the group of elephants had rushed into the stockade and there was no animal directly under the drop-gate, the gate-keeper (Figure 3c), stationed on top of the entrance, quickly cut the heavy jute rope, and the drop-gate fell to the ground, securing the elephants.

In general, the stockade team tried to take out captured elephants as soon as possible, normally within 24 h, because undue delays could cause additional injuries and stress. One elephant at a time was forced out of the stockade through an opening at the side of stockade (Figure 4) and moved to nearby “cradles” designed for breaking procedures (Figure 5). Old elephants, heavily pregnant elephants and adult bulls, desperately trying to free themselves and difficult to control, were released back into the wild.

## 7. Post-Capture Behavior Training (Taming)

All captive elephants in Myanmar were managed under a “free-contact” keeping system. It was of paramount importance that all captured elephants had submissive behavior and were responsive to handlers/mahouts. Elephants went through a taming procedure immediately after capture. Taming allowed the safe handling of elephants to alleviate the risk involved in the working environment for elephant handlers. The first step of taming was for the elephant to accept a human’s presence at her side. This included acclimatizing to human voices, tactile and visual contact and fire. Taming procedures involved a mixture of negative reinforcement (target behavior removes unpleasant stimuli), positive punishment (an unpleasant stimulus decreases unwanted behavior) and positive reinforcement (target behavior rewarded). It was normal for a calf or sub-adult to resist taming and refuse to eat in the first few days of the breaking-in process. Any injuries were treated to prevent infection. Drinking and feeding were arranged on a timely basis, whether they ate or refused to eat [45].

All taming procedures were usually started in the late evening under the light of a campfire. This practice has been criticized as sleep-deprivation, and thus cruel and brutal torture. However, the purpose of training by night was not to prevent sleep, but to reduce heat stress, accustom the elephant to fire, which was often used to drive wild animals, and more importantly, to carry out the process at a quiet time with few people around, thus allowing the elephant to selectively pick up the voices and commands of the men who would become their masters for the rest of their lives.

During the first few days, the group of trainers talked softly to the elephants or sang songs, known as shaw pike in Myanmar, while rubbing the rump, thighs, belly, chest, shoulders and other parts of the body to make touching and the voices of humans familiar. The mahouts then began to sit, stand or ride on the elephant’s back or neck (Figure 6). Once the elephant accepted the mahout sitting on its back, the first step of the training process was considered complete, and the elephant was classified as “broken” when it began to accept food, water and human contact/touch.

After the first signs of accepting a human riding on its back, the second step of taming aimed to habituate the elephant to a human-dominant society. The sequence of taming training during the second step was as follows:

Roping of the hind limbs was reduced, and the cradle (breast band) was removed. Elephants that had finished the intensive training program were allowed to learn to stroll in hobbles with a fettering chain attached to the hobble to prevent the animal from going astray.

The elephant was then frequently transferred to various locations around the training camp, to make it familiar with its terrain and habitat.

When the elephant became more docile and tractable, it was taken to lush pasture and allowed to graze by day, taken to a river for bathing and taken back to the camp at night. During this stage, the elephant was closely observed by the trainers to monitor its sleeping patterns and its ability to get up and walk comfortably with hobbles attached to the front limbs. Initially, trainee elephants in hobbles faced difficulties when rising from a reclining position. The trainers were instructed to observe the elephants around the clock and check for any signs of ill health, including taming-related injuries such as rope-burns, bruises, etc. The author, a vet herself, was on site to monitor the health condition of captured elephants. If necessary, the elephants were mildly sedated to calm their demeanor and administered antibiotics to prevent the infection of wounds.

Once the elephant was accustomed to the hobbles while sleeping, walking and rising, it was allowed to wander freely both day and night. Intensive training was continued, teaching simple commands, such as lift the leg, lower the front portion (kneel-down position), stop, trunk up, etc., which are indispensable for use as a working elephant. The duration of taming training usually lasted 3 to 8 weeks, depending on the learning ability of the individual elephant.

## 8. Identification of Government-Owned Elephants in Myanmar

After the completion of taming, each elephant is classified as a trained calf (TC) and assigned a permanent individual registration number (which is later branded on their rump as a permanent marking) (Figure 7). All government-owned elephants have two branding marks on each rump, with a star on top and the registration number below (Figure 7). Branding is performed using a corrosive paste, containing light kaolin, bentonite and caustic soda mixed with glycerin and methyl alcohol. Branding paste is applied to the clean, dried skin of the buttocks while tying the elephant’s tail under its belly and then leaving the elephant in the sun to hasten the drying of the paste. A swollen black mark appears within 24 h and, in due course, the skin peels off leaving a permanent scar of the branded markings after the routine application of dressing oil containing fly repellent and antibiotics.

After the completion of taming, each elephant is classified as a trained calf (TC) and assigned a permanent individual registration number (which is later branded on their rump as a permanent mark). Branding is also more easily viewed from a distance than ear tags and can be read at any time of the year. Then, an oozie is allocated to each elephant, along with a log book, known as “FORM J” in which its biodata (sex, temperament, musth, mating, calving, veterinary intervention, etc.) are recorded. This is usually conducted during the rest period in summer so that the working season is not disrupted.

A branding mark in the livestock industry is regarded as a property stamp identifying the owner of the animal. Elephant branding has been prohibited in recent times in many animal collections worldwide because of the physical trauma and psychological stress it causes. For animal welfare reasons, many veterinarians, animal welfare specialists and conservationists are globally promoting the method of implanting a microchip over the practice of branding for individual identification. Myanmar’s traditionally used branding of unique registration numbers on the elephant’s buttocks is permanent and relatively low in cost, and helps distinguish captive elephants from their wild counterparts, especially if they go astray in the forests. Elephants without visible marks for identification are logistically impossible to differentiate when they mix with their wild counterparts while foraging. With branding marks, captive elephants are easily distinguished from wild elephants, and the authority figure normally recaptures those elephants that fail to return to the base camps or willingly join the wild herd by noosing or by immobilization methods and retrieves them with the help of Konchee elephants. A noteworthy study in Germany stated that both the branding of and microchip implantations in horses induced an increase in cortisol release, but did not differ between the groups, concluding that the implantation of a microchip was as stressful as branding [46]. Myanmar still uses branding as ownership identification because branding is more easily viewed from a distance and the authorities responsible for wildlife trafficking and animal movements within the country are not normally provided access to the Myanmar elephant registration system, and most of them do not own microchip readers.

## 9. Discussion

The recently used method to capture wild elephants involves using tranquilizer darts. A specialized team, often including veterinarians, uses a dart (immobilization) gun to sedate the elephant. Drugs like etorphine (commonly known as M99) are used for darting wild elephants [47,48]. The dosage is carefully calculated based on the elephant’s size, body weight and health/body condition [49]

Here, I focus on the implications of elephant welfare. Live off-take, including selective removal from the wild by darting, trapping, noosing or netting, followed by handling, sampling, branding and attaching electronic devices, and so on, has significant welfare implications [50,51,52]. Capture always involves some risk of mortality, even in healthy animals, and can cause acute stress and injury [47]. It is usually more difficult as animal size increases, probably because of the increased difficulty of meeting their welfare requirements as well as safety precautions for the humans involved. Deaths may be directly attributable to the capture method itself (e.g., drug overdose, drowning or falling from high ground during the induction of anesthetics or during the initial phase of chasing/herding/driving, strangulation and/or asphyxiation by snares/rope used during capture) or may be caused by the secondary effects of capture (e.g., stress, infection, myopathy, trauma or instrumentation) and complications associated with sedative drugs used for immobilization [47,53].

Traditionally, the Kheddha system of capture targeted the capture of a group of elephants while the Milashkar (noosing) and immobilization methods captured singly selected individuals. While traditional, these methods faced criticism and ethical concerns due to the stress and potential harm and distress that could affect the elephants. According to the studbook data recorded for timber elephants by Burmese/Myanmar authorities for a century, captured wild elephants showed higher mortality than captive-born elephants, regardless of their capture method [54]. As discussed above, Myanmar elephant catchers generally targeted sub-adult elephants (with a shoulder height of ≈1.5 m or age of ≈10 y). Females were the preferred sex because the commonly held belief was that females lived longer and were easier to handle [10], but in the Kheddha system, the sex, age and size of animals could not be monitored due to the nature of the operation. For the capture of wild elephants by Milashikar (noosing) or darting, the chosen elephant was initially chased into a corner or forced to separate from the herd by a group of trained elephants. Similarly, in the Kheddah system described in this manuscript, the selected female group of elephants was driven through a diversion route leading to a stockade area. Chasing and cornering elephants for capture was a well-coordinated effort by skilled, professional elephant catchers. As an initial phase of herding or chasing is unfortunately unavoidable in many wild animal capture operations, it is not surprising that captured wild elephants showed similar results for long-term mortality and recovery rates at all ages, regardless of the capture method [54]

Elephants captured by the Kheddah system were generally older than those caught using the other two methods [32], and showed a higher mortality rate during the immediate post-capture period than elephants captured and tamed young [54]. Previous studies on MTE elephants documented that older elephants and males usually took longer to tame than younger elephants and were likely to suffer longer tame-related stress because of the greater risk of wound infection [33,55]. Taming or breaking is historically blamed for the post-capture mortality of Myanmar elephants [33,54,55]. In any wild capture scenario, mobility restriction, confinement and exposure to humans can cause physiological stress, and this varies with species, including age, sex, body weight or body/health condition, reproductive behavior/status, length of flight and time of chase [49]. When a wild animal is disturbed and herded unnaturally, it will instinctively flee for its life and run until it is exhausted. Anxiety and an instinctive or inborn fear of danger worsen the stress of animals, especially during the initial phase of chasing or driving and when the captives are in holding areas [47]. The traumatic experience coupled with social disruption and defeat after capture exaggerates post-traumatic stress [56].

Such psychological and physical-related stresses would be preventable if capture was performed with the aid of modern veterinary technology. Varieties of tranquilizers and anesthetic drugs are available nowadays to pacify wild animals and to induce standing sedation, especially those without any information on basic health data [49,57,58]. Standing sedation is different from deep sedation; it induces a total loss of consciousness and protective/defensive reflexes; and it is the state where an animal is not entirely unconscious and lying flat on the ground, but remains in a standing position with an awareness of its surroundings, though with less control of its movements, so wildlife personnel can safely intervene to restrain it and perform minor veterinary procedures with less or no resistance from the animal (details in [49,53]). A hand-made blow-pipe with an attached syringe is used for the administration of sedatives in short-range targets [58]. In the context of Kheddah, this would apply especially when the elephants were in the holding area of the stockade. Self-harm or physical injuries due to panic within the holding area, as well as the resistance of the captured elephants when they were taken out of the holding area of the stockade, is reduced if the captured elephants are administered systematically calculated sedated drugs via a blow-pipe. Hyperthermia, dehydration and capture myopathy are three leading causes of immediate post-capture mortality in wild captures [59]. Simple actions, such as hydration, cooling and sprinkling with water, can reduce the likelihood of hyperthermia and dehydration and can calm down the captured elephant. Early intervention with anti-inflammatory drugs and supportive care is critical to alleviating pain and stress [49]. Proper post-capture care with supplementary feeding and close monitoring with veterinary attention can improve survival rates.

Irrespective of how elephants were captured, mortality was high in the first year after capture, but decreased with time [54]. Unfamiliarity with the terrain and feeding grounds in captive environments could have caused inadequate feeding, leading to retarded growth and decreased immunocompetence in captured elephants, especially during the immediate post-capture period that coincided with summer. Although never documented in detail, one of the possible causes of post-capture mortality seemed to be restricted movement. After the end of the taming sessions, wild-caught elephants were trained to move with hobbles attached to their forefeet. Hobbling prevented the elephants from going astray, and the marks left on forest trails by the fettering chain attached to the hobbles guided the mahouts in tracing the whereabouts of the elephants for taking them back to the base camp after night foraging. Walking or trotting with hobbled forefeet in an unfamiliar terrain at night was energy-sapping for the newly tamed elephants. In general, the use of hobbles was discouraged. Normally, they were worn loosely enough to allow the elephant to walk without hopping when they were provided access to a natural habitat for foraging at night. However, the elephants trained to walk with hobbles could not compete with resident elephants in reaching the foraging ground in time, and the amount of leftover forage might not be enough to meet their dietary needs. Good-quality forage and water can be scarce in forests during summer (February to March), coinciding with post-capture hobbling training. Inadequate food and water intake could have lead to malnutrition and thus to reduced immune competence that later introduced the risk of complications from various infectious diseases, such as foot and mouth disease, anthrax, hemorrhagic septicemia, respiratory diseases, gastrointestinal complications and parasitism [32,60]. Although never mentioned anywhere in detail, malnutrition, in the summer months during the post-taming hobble training period, was likely the major factor enhancing the deleterious effect of perceived taming stress in newly tamed elephants. To avoid post-taming mortality, captured elephants should have been provided supplementary feeding before they were released for night foraging, especially in their first few years in captivity, so that they need not have wasted their resources on foraging, but only practiced getting used to walking with hobbles. Unfortunately, this practice of supplementation was not encouraged in those days.

Modern behavior training emphasizes positive reinforcement and animal welfare [61]. This leads to a more cooperative and trusting relationship between the animal and the trainer. Animal care strategies and training methodologies are a reflection of the increased knowledge of animal behaviors that allow the animals to make choices and control in their daily lives. The traditional taming method used in the past could be harsh, involving confinement and breaking the elephant’s will, which often relied on the dominance and physical control of the trainers [32,55]. However, Burmese elephant trainers believed that, once the taming was finished, a bond could be formed through the animal’s submission, often resulting in the animal becoming dependent on and obedient to the human for direction and care, and they respond to and behave well with the mahouts they have known for a long time [62]. A good or positive human–animal relationship is regarded as bonding, and it provides welfare benefits to the animals [63]. Advances in animal welfare science and a deeper understanding of the basic needs of elephants in captivity have enabled methods for taming elephants to be fine-tuned over the past decade. Positive-reinforcement training, known as PRT, is being increasingly promoted throughout Asia, including Myanmar, and it is now used for cooperation after an initial traditional taming period [55,62,64] Timber camps and ecotourism camps are now focusing on behavioral enrichment, providing activities and environments that stimulate animals mentally and physically, encouraging natural behaviors. If Kheddah is practiced in Myanmar, the survival outcome of captured elephants will be better than the operations in the early 1990s.

Wild capture can hinder long-term conservation goals by reducing remaining wild populations, but it can also be viewed as restoring the gene flow to recently fragmented captive populations. The historical data of MTE elephants in Myanmar show that capture from the wild has lasting adverse effects on lifetime reproduction [65] with increased long-term mortality in both males and females [54]. Elephants captured and tamed at older ages show a higher increase in mortality after capture than elephants captured and tamed young. The Khedda [55,62,64] system presented in this manuscript targeted the female group. Although it is not possible to choose the age of the animal, the post-capture mortality of older elephants can be reduced by the immediate release of unwanted old elephants. The author is not suggesting that reinstating the capture of wild elephants is a potential solution to replenish the captive elephant population, and instead, would like to suggest that management should focus on the breeding of the captive elephants.

## 10. Conclusions

The practice of capturing and training wild elephants in Myanmar is a complex issue, deeply embedded in the country’s history, economy and forest management. Elephants have been taken from the wild to replenish the captive stock since Burma was under British rule, simply because of the unavailability of captive breeding programs in those days. Wild off-take and the use of Kheddah for capturing wild elephants have been officially banned in Myanmar since 1985, and have never been reinstated. Still, the author wishes to pass on to today’s conservationists and personnel involved in elephant management the details of the Kheddah method, which was used as a viable tool for the emergency removal of locally overabundant elephant populations in Myanmar, when the capture of wild elephants using an individually selected capture method, such as immobilization or Milashikar, is not practical. There are possibilities to reduce the immediate or long-term post-capture physical and psychological stresses in captured elephants with the help of modern veterinary procedures, science-based management practices and systematic behavior training. However, as ethical considerations and conservation efforts are gaining momentum, there is a gradual shift toward more sustainable and humane practices toward elephants in captivity. The future of Myanmar’s elephants will hopefully depend on balancing traditional practices with modern conservation needs.

## Figures and Tables

**Figure 1 animals-14-02506-f001:**
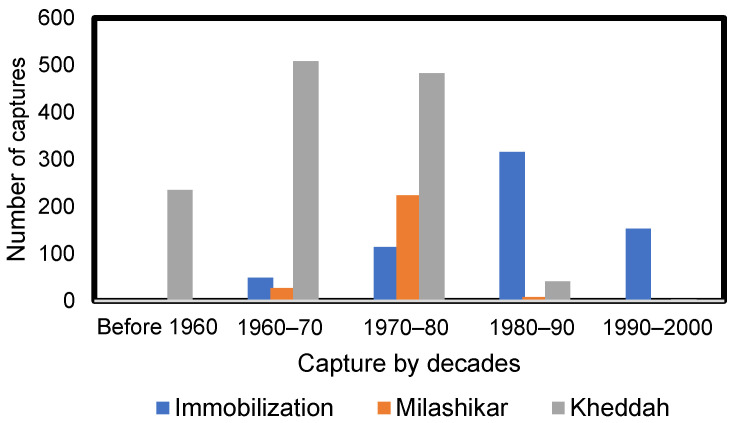
Wild-caught elephants captured by decade and by capture method, from before 1960s to the end of 2000 [32].

**Figure 2 animals-14-02506-f002:**
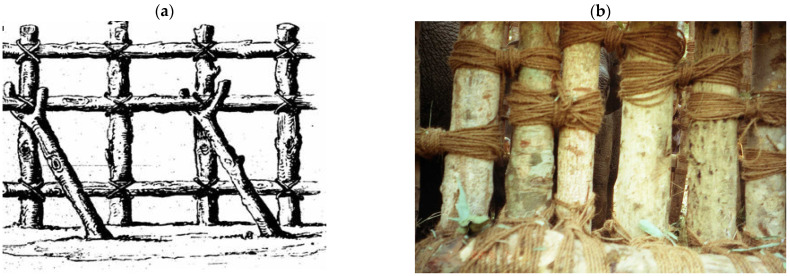
(**a**) Detail of stockade construction [42]. (**b**) The fences are elephant-proof constructions fastened with strong strands of ropes. Photo by author.

**Figure 3 animals-14-02506-f003:**
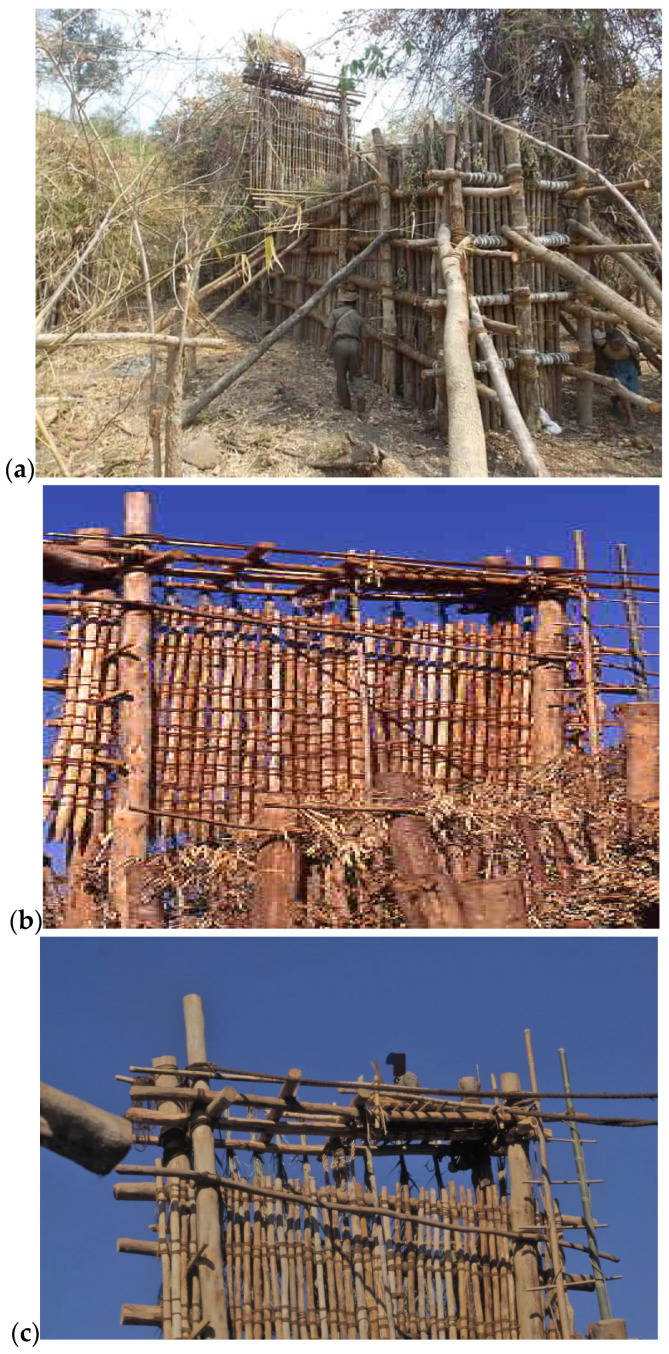
(**a**) General structure of a stockade built in 1996: the view from outside the dead end, (**b**) the drop-gate heavily camouflaged by leafy branches and (**c**) the gate-keeper on top of the drop-gate. Photos by author.

**Figure 4 animals-14-02506-f004:**
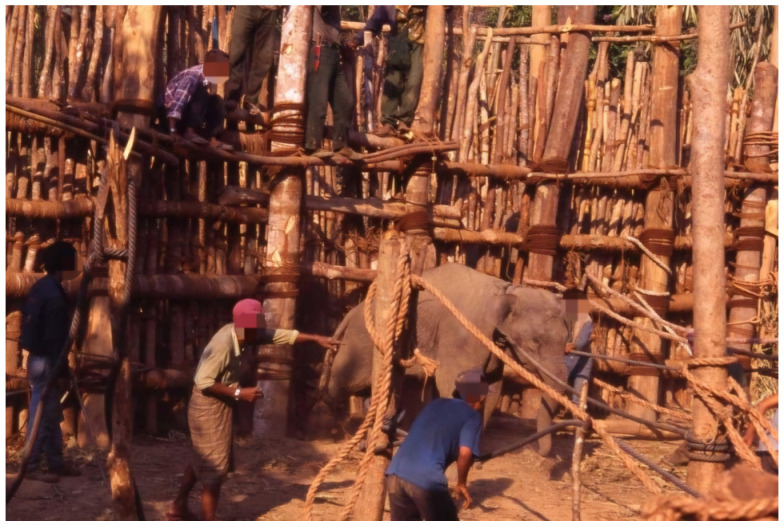
One elephant is taken out at a time from the opening at the side of the holding area of the stockade using manpower. Photo by author.

**Figure 5 animals-14-02506-f005:**
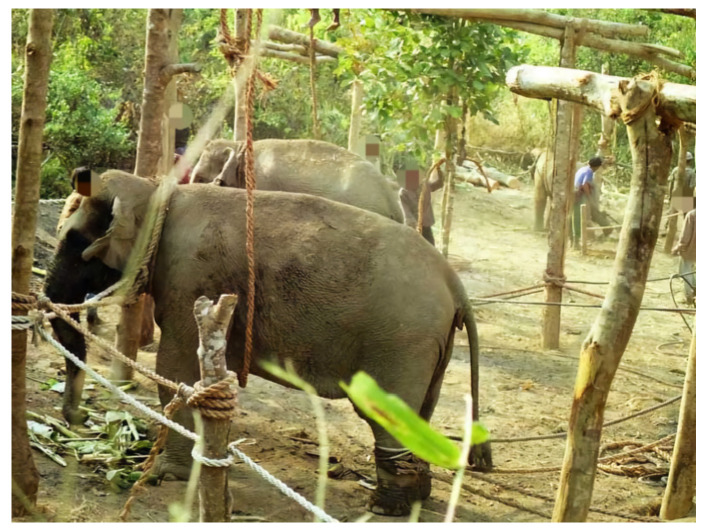
Taming of all-female groups follows immediately after capture. Photo by author.

**Figure 6 animals-14-02506-f006:**
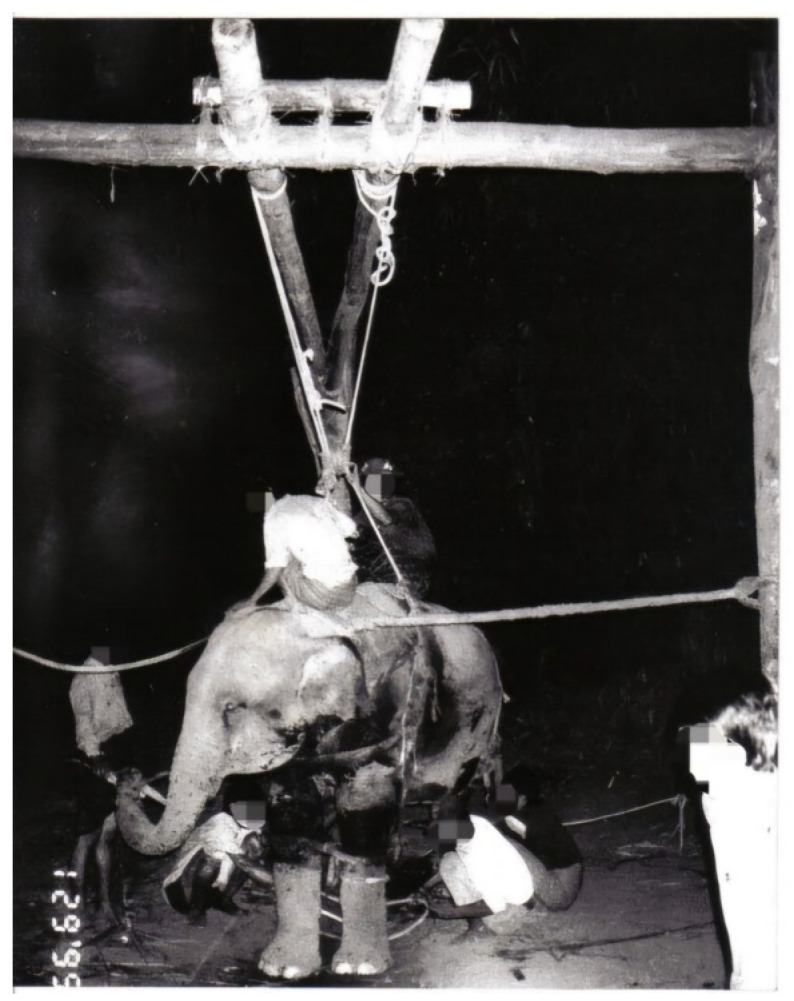
Taming using a cradle and a mahout on the elephant’s back. Photo by author.

**Figure 7 animals-14-02506-f007:**
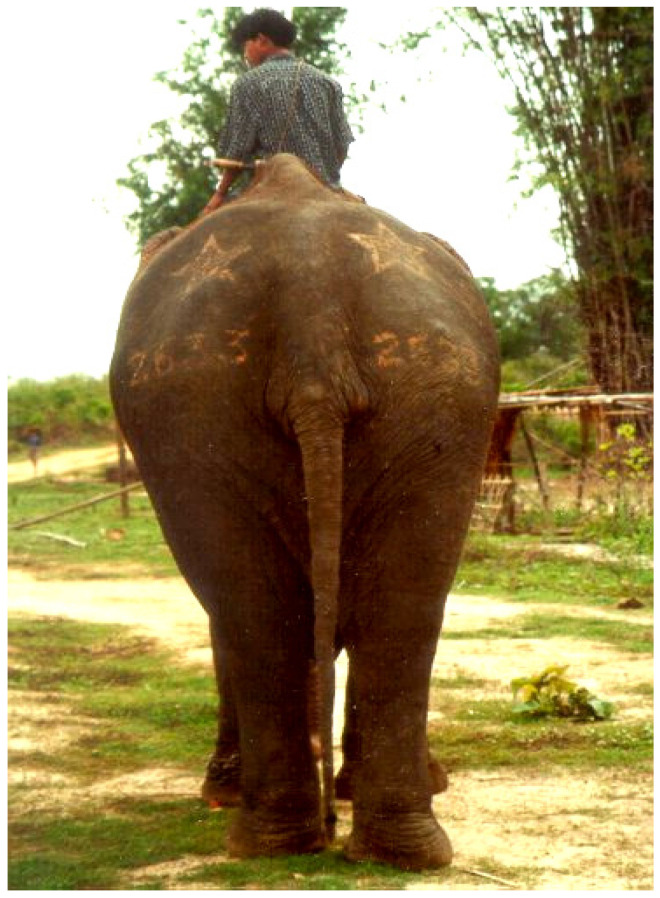
Timber elephant with its registration number and a star, branded on its rump (a star above the numbers denotes government-owned elephants). Photo by author.

## Data Availability

The data are available from the corresponding author with reasonable request.

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
