# Peer review of "Historical Account of Managing Overabundant Wild Asian Elephants in Myanmar by the Kheddah System of Capture: Philosophy, Principles and Practices"

_animals, 2024, doi:10.3390/ani14172506_

Round 1

Reviewer 1 Report

Comments and Suggestions for Authors

The article provides a valuable historical record of a method for capturing wild elephant in Myanmar, which has not been previously published due to its confidential nature. This novel contribution is appreciated. However, I have some concerns and suggestions to help improve the clarity and focus of this article.

General comments:

-              Please note that it is important and very crucial to emphasize that the capturing method described does not meet current welfare standards and should not be suggested as applicable to present situation. This article is more likely for historical and educational purposes, please considering edit the title as well.  

-              I notice several repetitive data/statements in your article and also the tone of your article leans towards being quite informal. I recommend revising the overall structure/language for clarity and conciseness.

Specific comments:

1. Introduction: it was briefly mentioned about the importance of wild elephants in the ecosystem and addressed the declining of wild elephant populations, which require urgent attention. However, your article was described about the capturing elephant out from the wild, which worsen the wild elephant population. This creates confusion about the article introduction, objectives, and relevance to the conservation efforts. Please consider rewrite your introduction part. 

2. About Myanmar: consider changing to topic to Myanmar Forest Management System, as this part talked about the forest management only. 

3. Logging elephants of Myanmar

                  - Line 118: please add reference to “Myanmar is widely recognized as home to the largest captive population of approximately 6000 elephants.”

                  - First and second paragraph seems not connected, confusing. 

                  - Line 133: what is “calves-at-heel, old”?

                  - Line 133: female with pregnancies (>55yr) – can a female aged more than 55 yr still become pregnant? Please rewrite the sentence for clarification. 

4. Background history of wild elephant capture: I found that this part of your article is very hard to follow – suggesting to use a subtopic, avoid minor/extraneous details which can distract reader from the main points, adding info-graphic about the timeline may also help. 

5. Philosophy of Kheddah: Figure 1 should be moved and merged with topic 4 as it described about each method of wild elephant capture. This topic is actually could combine with the next topic (principle of kheddah)

6. Principle of Kheddah:

                  - Line 372: why did Figure 6 come before Figure 2?

                  - Line 397-399: recommend to draw a new figure that indicate the length or put a length in the existing figure.

                  - Figure 2 (a): please change this figure due to the poor quality

                  - Figure 2 caption: (c) should be put in front of the stated sentence. Please make sure the writing style of each figure caption are similar. 

8. Post-capture behaviour training 

                  - Line 505-509: repeated data with previous paragraph.    

                  - Is there any welfare/health monitoring during taming session?

9. Identification:

-              Fig 7b was not good in quality, may be removed?

-              Line 567-568: "Branding and microchipping would not be different in causing stress, based on the study on horse". This should be very carefully discussed as the size of branding in elephant in Myanmar was very large when comparing to the tiny microchip, thus I think this issue could not compare with the study in horses, and branding in elephant should not be recommend based on welfare and trauma issue. 

-              Lin 575-onward: this part should be moved as it’s not related to the identification.

Discussion: I strongly encourage the authors to added and discussed more about the current capturing methods (pros and cons) comparing to the Kheddah. Also, if we want to apply the Kheddah to the current situation, which welfare issue (or which step) can be improved – probably adding data on the last Kheddah that the author in charge of in 1996, was there any improve in welfare comparing to the formal Kheddah?

References: Please carefully check your consistency of reference style (e.g., some reference used full name of Journal and some used abbreviation). 

Author Response

Thank you very much for your wise suggestions and recognize my work as a valuable historical record .  Repetitive statements are edited as suggested are corrected ; the whole manuscript is shortened and I try to be concise throughout. My answers is in bold and italic.

Comment (1). Introduction: it was briefly mentioned about the importance of wild elephants in the ecosystem and addressed the declining of wild elephant populations, which require urgent attention. However, your article was described about the capturing elephant out from the wild, which worsen the wild elephant population. This creates confusion about the article introduction, objectives, and relevance to the conservation efforts. Please consider rewrite your introduction part. 

I rewrite the Introduction and explain the current conservation status of elephants in Myanmar  in line 76-81 as that (despite the significant removals of wild elephants for logging operations, wild elephants thrived well and the population remained high into the 1970s and a range-wide assessment of the remaining elephants in 2003 suggested that Myanmar/Burma still possessed more quality habitat than neighbouring countries)

Comment 2. About Myanmar: consider changing to topic to Myanmar Forest Management System, as this part talked about the forest management only. 

Paragraph 2 is subtitled as Utilization of captive elephants in Myanmar, combining the two paragraphs of  About Myanmar and Logging elephants of Myanmar of my previous version of manuscript

Comment 3. Logging elephants of Myanmar

                  - Line 118: please add reference to “Myanmar is widely recognized as home to the largest captive population of approximately 6000 elephants.”

References added in  (line 125-126)

                  - First and second paragraph seems not connected, confusing

It is corrected and explained in one para (line124-134 in my new version)

                  - Line 133: what is “calves-at-heel, old”?

Calves under 5 year old are classified as calves-at- heel (CAH) according to Myanmar way of classification

                  - Line 133: female with pregnancies (>55yr) – can a female aged more than 55 yr still become pregnant? Please rewrite the sentence for clarification. 

It is corrected (in lines 138-139). I rewrite as follows.

Only half of the MTE elephant population (≈ 1600) can engage in harvesting operations at any one time, because young elephants (< 18y), pregnant females, females with calves (<5y), classified as calves-at-heel (CAH), old elephants (>55y) and sick elephants are not allowed to work in extraction operations (Aung and Nyunt, 2002a)

Comment 4. Background history of wild elephant capture: I found that this part of your article is very hard to follow – suggesting to use a subtopic, avoid minor/extraneous details which can distract reader from the main points, adding info-graphic about the timeline may also help. 

This paragraph is no 3. Capture of wild elephants, removing the details of historical capture schemes

Comment 5. Philosophy of Kheddah: Figure 1 should be moved and merged with topic 4 as it described about each method of wild elephant capture. This topic is actually could combine with the next topic (principle of kheddah)

This chapter is now Para 4. I agree with you. But I want to explain Government arrangement of capture in the early set-up of MTE and how MTE chooses private catchers to capture wild elephants.

  1. Principle of Kheddah:

                  - Line 372: why did Figure 6 come before Figure 2?

Apologies for putting Figure 6 in wrong place; remove Figure 6 (line 270)

                  - Line 397-399: recommend to draw a new figure that indicate the length or put a length in the existing figure.

I am sorry to draw new sketch here because of the restricted time I have to finalize this manuscript

                  - Figure 2 (a): please change this figure due to the poor quality

Remove because I can't find better photos in my collection of photos

                  - Figure 2 caption: (c) should be put in front of the stated sentence. Please make sure the writing style of each figure caption are similar.                                                                                             Correct as suggested

  1. Post-capture behaviour training 

                  - Line 505-509: repeated data with previous paragraph.    

Noted with thanks and removed

                  - Is there any welfare/health monitoring during taming session?

  I add information of health monitoring in Lines 422 to 426 as below-

  • The trainers were instructed to observe the elephants around the clock and check for any signs of ill health including taming –related injuries such as rope-burns, bruises, etc. The author, a vet herself was on-site to monitor the health condition of captured. If necessary, elephants were mildly sedated to calm their demeanour and gave antibiotics to prevent infection of wounds.

Comment-9. Identification:

-              Fig 7b was not good in quality, may be removed?

Remove as suggested

Comment 10-              Line 567-568: "Branding and microchipping would not be different in causing stress, based on the study on horse". This should be very carefully discussed as the size of branding in elephant in Myanmar was very large when comparing to the tiny microchip, thus I think this issue could not compare with the study in horses, and branding in elephant should not be recommend based on welfare and trauma issue. 

Noted with thanks- I try to explain that  Myanmar cannot use milder method of animal identification method such as microchipping because because branding is more easily viewed from a distance and authority responsible for wildlife trafficking and animal movement within the country are not normally given access to Myanmar elephant registration system and most of them do not own microchip readers (Lines 466 to 469).

Comment 11.   Lin 575-onward: this part should be moved as it’s not related to the identification.

 Agree and remove lines 575 to 596 of my previous manuscript

Comment 12- Discussion: I strongly encourage the authors to added and discussed more about the current capturing methods (pros and cons) comparing to the Kheddah. Also, if we want to apply the Kheddah to the current situation, which welfare issue (or which step) can be improved – probably adding data on the last Kheddah that the author in charge of in 1996, was there any improve in welfare comparing to the formal Kheddah?

 Rewrite the whole Discussion, incorporating your wise suggestions 

Comment 12 --References: Please carefully check your consistency of reference style (e.g., some reference used full name of Journal and some used abbreviation). 

Thank you for pointing out my errors. My Endnote version needs to readjust.

Reviewer 2 Report

Comments and Suggestions for Authors

I see the benefits of this paper in that it explains and defines something that is probably controversial and requires management or regulation. The paper is long and too wordy in places. I would recommend that it is cut down by a 1/4 to encourage people to read it through. The style of sentences can be choppy and hard to follow. There are strange uses of font style, size and emphasis (e.g., underline or bold). This needs to be written as an objective and unbiased piece of academic prose, and not as a journalistic or opinion piece. Please check and edit throughout. 

In its current form, I cannot recommend this article for publication. But with a thorough edit to reduce words, tighten up meaning, and provide better explanation and evaluation of key concepts, this would be useful.

I recommend a structure of:

1. Introduction (to elephants, their management, and specifically those in Myanmar)

2. Logging elephants and human issues 

3. The Kheddah System and its implications on elephant welfare

4. What improvements, developments or changes are needed to promote elephant welfare

5. General dicussion

6. Conclucions.

Ensure that each and every point taken from the outside literature is cited. Do not assume the reader has technical knowledge so make sure you explain and describe key words or technical language. 

The referencing format used is incorrect for the journal. This should be numbered and an APA style has been used in the text.

Should Kheddah system be in " "? As it's a given name? 

Kheddah method or Kheddah system? This are used interchangeably and I would stick to one to be consistent. 

The abstract is very introduction heavy and does not provide a summary of what has been found from the review. Also, the author needs to explain why they are not endorsing the system, otherwise this appears to be a "throw-away" comment.

Line 45: incorrect citations for the Asian elephant's assessment on the IUCN Red List page. Given that the Red List page for each species actually tells you how to cite it, this is pretty sloppy.

Keystone and umbrella species need both explanation and citations. 

The same is true for flagship species.

Why are some text in larger font, underlined or in bold?

The introduction needs better citations and a more fluent writing style. The section "About Myanmar" should be included as a subheading of the introduction. 

Under section 3, some of this information on Asian elephant biology belongs in the introduction. Then the focus can be on logging elephants specifically. 

Where do these data in Figure 1 come from? Please evaluate this figure in the text. 

Do the photos require permissions or copyright? Please can these be explained in the text as to what they show and why they are needed?

The conclusion reads more like an abstract rather than a final summary of what has been found. Please re-write the conclusion so that you give two key concepts; 1) the overall findings or results from the research and 2) what the wider importance or take-home message of these is. 

Comments on the Quality of English Language

This needs to be improved to make the discussion and evaluation more succinct and easy to follow. There needs to be better bridges between concepts to improve the narrative of the paper.

Author Response

 I am very grateful for your guidance.

I have edited and shortened the manuscript as per your suggestions. I incorporate your wise suggestions in my new version of the manuscript. My answers are in bold and italic

Comment 1. Introduction (to elephants, their management, and specifically those in Myanmar)

I rewrite the whole introduction as suggested

Comment 2. Logging elephants and human issues

This is mentioned in one paragraph as Para 2 - Utilization of elephants in Myanmar 

Comment 3. The Kheddah System and its implications on elephant welfare

It is discussed in length in the Discussion

Comment 4. What improvements, developments or changes are needed to promote elephant welfare

In the discussion, I explain that post-capture mortality of the Kheddah system would be preventable if it is done with the assistance of  modern veterinary intervention, animal care strategies and training methodologies based on positive reinforcement training. I explained in length that if Kheddah is practiced  nowadays, the survival outcome of captured elephants will better than those operations in the early 1990s

Comment 5. General discussion

Comment 6. Conclucions.

In Conclusion, I focus on the welfare implications of Kheddah system of capture and conclusion is rewritten to the best of my knowledge.

Comment 7. Ensure that each and every point taken from the outside literature is cited. Do not assume the reader has technical knowledge so make sure you explain and describe key words or technical language. 

Thank you. I make sure my new version has appropriate citations

Comment 8. The referencing format used is incorrect for the journal. This should be numbered and an APA style has been used in the text.

I follow your suggestions

Comment 8. Should Kheddah system be in " "? As it's a given name? 

In old literatures by(Sanderson, 1878; 1883; Buckland, 1887), Stockade capture of wild elephant was mentioned as Kheddah without apostrophe

Buckland, C., 1887. Elephant-Hunting in India. Longman's magazine, 1882-1905 11, 37-45.

Sanderson, G., 1878. Thirteen years among the wild beasts of India; Their hunts and habits from personal observation with an account of the modes of capturing and taming wild elephants. WH Allen, London. Elephant catching, 361-382.

Sanderson, G., 1883. Pack gear for elephants. The Suprintendent of Government Printing, India, Culcutta.

Comment 8. Kheddah method or Kheddah system? This are used interchangeably and I would stick to one to be consistent. 

I use the wording Kheddah system in my manuscript, as in old literatures

Comment 9. The abstract is very introduction heavy and does not provide a summary of what has been found from the review. Also, the author needs to explain why they are not endorsing the system, otherwise this appears to be a "throw-away" comment.

Thank you. The introduction is rewritten and I explain in length in Discussion that If Kheddah is practiced in Myanmar nowadays, the survival outcome of captured elephants will better than those operations in the early 1990s. Kheddah mortality was high in the old days simply because of the lack of updated knowledge on veterinary intervention and failure to fulfil the much-needed animal welfare requirements of captured wild elephants

Comment 10. Line 45: incorrect citations for the Asian elephant's assessment on the IUCN Red List page. Given that the Red List page for each species actually tells you how to cite it, this is pretty sloppy. Keystone and umbrella species need both explanation and citations. The same is true for flagship species.

Rewritten as per respective web sites and with proper citations

Comment 11. Why are some text in larger font, underlined or in bold?

The font-sizes of whole document has been adjusted uniformly in the new version of manuscript

Comment 12. The introduction needs better citations and a more fluent writing style. The section "About Myanmar" should be included as a subheading of the introduction. 

Paragraph 2 (About Myanmar) and paragraph 3 (Logging elephants of Myanmar) are combined to form one paragraph (Paragraph 3) titled “Utilization of captive elephants in Myanmar”

Comment 13. Under section 3, some of this information on Asian elephant biology belongs in the introduction. Then the focus can be on logging elephants specifically. 

Yes, in the new version, I mention in  the Asian elephant biolgy in (Paragraph 3- Utilization of captive elephants in Myanmar)

Comment 14. Where do these data in Figure 1 come from? Please evaluate this figure in the text. 

These data are extracted from my Thesis (Mar, K.U., 2007. The Demography and Life-history Strategies of Timber Elephants of Myanmar, PhD Thesis, University College of London, London, UK). The data is now attached as a supplementary table

Comment 15. Do the photos require permissions or copyright? Please can these be explained in the text as to what they show and why they are needed?

All photos are taken by myself and explain in the text as well as in documentary film

Comment 16. The conclusion reads more like an abstract rather than a final summary of what has been found. Please re-write the conclusion so that you give two key concepts; 1) the overall findings or results from the research and 2) what the wider importance or take-home message of these is. 

The conclusion is rewritten as follows

Conclusion

The practice of capturing and training wild elephants in Myanmar is a complex issue, deeply embedded in the country’s history, economy and forest management. Elephants have been taken from the wild to replenish the captive stock since Burma was under British rule,  simply because of the unavailability of captive breeding programs in those days. Wild off-take and the use of Kheddah for capturing wild elephants have now been officially banned in Myanmar since 1985 and were never reinstated. Still, the author wishes to pass on today’s conservationists and personnel involved in elephant management the details of the Kheddah method that was used as a viable tool for emergency removal of locally overabundant elephant populations in Myanmar when the capture of an individually selected capture method of wild elephants by immobilization or Milashikar was not practical. There are possibilities to reduce the immediate or long-term post-capture physical and psychological stresses in captured elephants with the help of modern veterinary procedures, science-based management practices and systematic behavior training. However, as ethical considerations and conservation efforts are gaining momentum, there is a gradual shift toward more sustainable and humane practices of elephants in captivity. The future of Myanmar's elephants will hopefully depend on balancing traditional practices with modern conservation needs.

Regarding Comments on the Quality of English Language- I have written the manuscript with the best of my English. If needed I will ask a professional English editing service for correctness and clarity to improve the narrative of the paper.

Round 2

Reviewer 1 Report

Comments and Suggestions for Authors

The revised manuscript has significantly improved after the revision, reflecting the author's careful attention to detail. I have no further concerns with the content of the manuscript. However, the English language should be thoroughly reviewed for grammar, typographical errors, and adherence to academic and scientific writing standards. Also, the references should be rechecked for typo error and consistency in formatting.  

Author Response

Dear Sir,

I greatly appreciate your kind approval of my paper.

As you suggested, I tried my best to recheck the errors and typos in my text and the uniformity of my reference list.

I added and rephrased some lines in Section 9 (Discussion). The remaining texts are untouched. I was suggested to shorten the Discussion.  The highlighted lines are the ones I added to this version.  

I hope you agree with the revised version of my paper.

Yours sincerely

Khyne Mar

Reviewer 2 Report

Comments and Suggestions for Authors

This is much improved with a clear flow and a narrative that brings the reader through the topic and the challenges of elephant management. The edits are clear and the examples well presented. I have one final suggested edit, which is that the paper is long. Therefore, I recommend shortening the discussion (section 9) to salient points that put context and critique onto the information presented in sections 1 to 8. Don't include new information in this discussion section, but summarise, evaluate, explain and provide the context for the information presented earlier in the paper. This will make the paper shorter, provide a more concise review of your main concepts and therefore mean people are more likely to read the whole article.

I am happy to recommend this paper for publication if the discussion is edited and reviewed as suggested, i.e., the discussion becomes a synopsis of the key information presented in sections 1 through to 8 and put into context. 

Comments on the Quality of English Language

Please give the article a thorough proof read and check sentence structure. Otherwise, quality of written English is good in parts. 

Author Response

I am very grateful for your suggestions to improve my manuscript and receive your approval for publication.

I have made changes in Section 9 as per your suggestion, and I am confident that these revisions have strengthened the manuscript. The remaining texts are untouched.   I have shortened the section 9 and the parts I re-worded and rephrased are highlighted in yellow.  Per your suggestions, new information not mentioned in the previous sections has been deleted. I hope to receive your approval  for publication

Yours sincerely,

Khyne Mar